# White Lupine (*Lupinus albus* L.) Flours for Healthy Wheat Breads: Rheological Properties of Dough and the Bread Quality

**DOI:** 10.3390/foods12081645

**Published:** 2023-04-14

**Authors:** Luciano M. Guardianelli, Bruna Carbas, Carla Brites, María C. Puppo, María V. Salinas

**Affiliations:** 1Centro de Investigación y Desarrollo en Criotecnología de Alimentos (CIDCA), Facultad de Ciencias Exactas-UNLP-CONICET, 47 y 116, La Plata 1900, Argentina; 2National Institute for Agricultural and Veterinary Research (INIAV), I.P., Av. Da República, Quinta do Marquês, 2780-157 Oeiras, Portugal; 3Centre for the Research and Technology of Agro-Environmental and Biological Sciences, University of Trás-os-Montes and Alto Douro (CITAB-UTAD), 5000-801 Vila Real, Portugal; 4GREEN-IT Bioresources for Sustainability, ITQB NOVA, Av. da República, 2780-157 Oeiras, Portugal; 5Facultad de Ciencias Agrarias y Forestales, Universidad Nacional de La Plata, 60 y 119, La Plata 1900, Argentina

**Keywords:** lupine, dough rheology, baking quality, sustainable protein

## Abstract

Protein-based foods based on sweet lupine are gaining the attention of industry and consumers on account of their being one of the legumes with the highest content of proteins (28–48%). Our objective was to study the thermal properties of two lupine flours (Misak and Rumbo) and the influence of different amounts of lupine flour (0, 10, 20 and 30%) incorporations on the hydration and rheological properties of dough and bread quality. The thermograms of both lupine flours showed three peaks at 77–78 °C, 88–89 °C and 104–105 °C, corresponding to 2S, 7S and 11S globulins, respectively. For Misak flour, higher energy was needed to denature proteins in contrast to Rumbo flour, which may be due to its higher protein amount (50.7% vs. 34.2%). The water absorption of dough with 10% lupine flour was lower than the control, while higher values were obtained for dough with 20% and 30% lupine flour. In contrast, the hardness and adhesiveness of the dough were higher with 10 and 20% lupine flour, but for 30%, these values were lower than the control. However, no differences were observed for G′, G″ and tan δ parameters between dough. In breads, the protein content increased ~46% with the maximum level of lupine flour, from 7.27% in wheat bread to 13.55% in bread with 30% Rumbo flour. Analyzing texture parameters, the chewiness and firmness increased with incorporations of lupine flour with respect to the control sample while the elasticity decreased, and no differences were observed for specific volume. It can be concluded that breads of good technological quality and high protein content could be obtained by the inclusion of lupine flours in wheat flour. Therefore, our study highlights the great technological aptitude and the high nutritional value of lupine flours as ingredients for the breadmaking food industry.

## 1. Introduction

The white lupine (*Lupinus albus* L.) is a legume native to the Mediterranean region and North Africa [1] and has been cultivated since ancient times for various uses, including fodder, fertilizer and seed consumption [2]. Chile is a high producer of the flour obtained from this nutritionally seed [3]. The seeds or beans are known by different names: lupines or lupinos in Latin America, altramuz in Spain, tremoço in Portugal and Brazil and lupins in English-speaking countries. Lupine is currently attracting a lot of interest because of its nutritional value [4] and its ability to adapt to poor soils, competing with soybeans and other legumes [5,6]. Regarding the composition of lupine, it has been found that it has double the protein amount of other legumes consumed by humans. The protein content varies between 28% and 48% according to the species, growing conditions and type of soil [7,8]. Globulins represent 80–90% of the total proteins and are high-molecular-weight storage proteins. Based on their electrophoretic mobility, globulins can subdivide into α-conglutin (11S, 35–37% of the total globulins), ß-conglutin (7S, 44–45%), γ-conglutin (10S, 4–5%) and δ-conglutin (2S, 10–12%) [9]. Albumin represents 15% of total proteins, while glutelins and prolamines are found in a lower proportion compared to other legumes [10]. In addition, lupine is a good source of essential amino acids; although the content of sulfur amino acids is low, it has a high content of lysine [11]. Although lupine is not considered an oilseed, it has a non-negligible content of crude oil (up to 15%), with an adequate balance of fatty acids: 10% saturated and 90% unsaturated fatty acids, including (18:1) oleic, (18:2) linoleic and (18:3) linoleic acids [12,13,14]. In addition, lupine is a good source of minerals, among which are potassium, calcium, magnesium and sodium [15]. Other molecules, such as the antioxidant phytochemicals in lupine, are known to have health benefits such as the prevention of various diseases associated with oxidative stress, such as cancer, cardiovascular disease, neurodegeneration, and diabetes [16]. On the other hand, lupine flour has caused allergies with the increase in consumption of this legume, which were reversed by heat treatment, especially in an autoclave at 138 °C for 20 min [17], without compromising the technological characteristics of the doughs and breads [18]. The demand for plant-based protein foods is expanding and presents an opportunity to satisfy the nutritional needs of the world’s growing population while transitioning to more sustainable food production [6]. Therefore, the development of high-quality foods enriched with dietary protein are important for industry, consumers and government institutions. On the other hand, bread plays a very important role in the human diet since relatively large quantities are consumed around the world [19]; therefore, is an interesting matrix for the incorporation of vegetable protein ingredients. An increase of 1.43% between 2019 and 2024 in world bread consumption was reported due to its convenience, portability, nutrition and taste [20]. Moreover, recently, consumer demand for healthy bread, mainly those enriched in the amount of protein and fiber, has increased [21]. For that reason, the purpose of this research is to enhance the nutritional value of breads with lupine flours. The objectives of this work were (1) to assess the flours’ thermal properties; (2) to assess the hydration and rheological behavior of the dough made with wheat flour complemented with flours obtained from seeds of two cultivars (Misak and Rumbo) of white lupine; and (3) to evaluate the baking quality of the breads.

## 2. Materials and Methods

### 2.1. Samples

Commercial wheat flour type 0000 (Molino Campodónico Ltd., La Plata, Buenos Aires, Argentina) [22] suitable for breadmaking with 14.3% of protein was used;White lupine (*Lupinus albus* L.) flour from the Portuguese cultivar, Misak, and the cultivar from Chile, Rumbo, were used.

### 2.2. Percentage Composition of Lupine Flours

The protein content was studied by Kjeldahl methods and lipids by Soxhlet. Moreover, ash and moisture content were determined according AACC methods [23].

### 2.3. Thermal Properties of Lupine Flour Suspension

The thermal properties of the lupine flours (Misak and Rumbo) were determined by differential scanning calorimetry (DSC) using a Q100 equipment (TA Instruments, New Castle, England). About 15 mg of flour suspension (40% *w*/*v*) in distiller water prepared 24 h before analysis was placed into aluminum capsules which were hermetically sealed and subjected to one heating cycle. Suspensions were heated from 5 °C to 140 °C at a rate of 5 °C/min [24]. Protein denaturation was characterized by different temperatures: onset (T0), peak (Tp) and final (Tf). The enthalpy associated to protein denaturation (ΔH) was determined between T0 and Tf. All determinations were analyzed by duplicate.

### 2.4. Hydration and Rheological Properties of Dough

Blends of 100 g of wheat flour with the addition of different levels (10, 20 and 30%) of lupine Misak (M10, M20, M30) or Rumbo (R10, R20, R30) were prepared. A wheat sample without lupine flour was included as control (C) in both cases. Moreover, 2% of NaCl was added.

#### 2.4.1. Farinograph Assays of Blends

Water absorption (Wabs), development time (td), stability (St) and softening degree (SD) of the different lupine wheat blends were determined using a 300 g-Brabender farinograph (Duisburg, Germany) according to AACC methods [23].

#### 2.4.2. Dough Formulation

Solid ingredients were mixed in a planetary small-scale kneader (Kenwood Major, Milano, Italy) for one minute at 50 rpm. Then water (according to Wabs values) was added to the solid blend. The mix was first kneaded at 50 rpm (speed 1) for 1 min; for the rest of the time, until reaching td, it was mixed at 90 rpm (speed 2). Final dough temperature was 24 ± 1 °C. Dough was laminated four times, rested 15 min at 25 °C, and covered with a plastic film for avoiding loss of water. Hydration and rheological trials on dough were performed without yeast. Doughs were performed by duplicate.

#### 2.4.3. Moisture and Water Activity

Moisture content of dough was determined according to AACC Method 44-19 [23] by dehydration in a stove (San Jor, Buenos Aires, Argentina). Water activity was measured using a Meter Aqualab series 3 (Decagon Devices Inc., Washington, DC, USA). Assays were performed in duplicate (*n* = 4).

#### 2.4.4. Molecular Mobility

The molecular mobility of the different dough was determined through relaxation assays using NMR Brüker Minispec equipment (Brüker, Billerica, MA, USA). A portion of dough was placed in a glass tube (diameter: 10 mm, height: 30 mm) that was closed to avoid dehydration. Intensity of ^1^H signals were recorded during time. Nuclei are excited for a few milliseconds and when the pulse stops, they return to the ground state emitting a signal. Relaxation curves of the proton (^1^H) signaling intensity versus time were fitted to a one-term exponential model according to Equation (1) and the ^1^H spin–spin relaxation times (λ) were determined using the Carr–Purcell–Meiboom–Gill pulse sequence according to Salinas et al. [25]:I(t) = A exp (−t/λ)(1)
where I(t) represents the ^1^H signal intensity, t is the time, λ is the relaxation time (a constant parameter) and A is the signal intensity of protons at t = 0. Assays were performed in duplicate (*n* = 4).

#### 2.4.5. Dough Texture

Dough was laminated (thickness = 1 cm) before cutting. Cylindrical dough pieces (diameter = 3 cm) were cut with a cylindrical aluminum puncher. Dough texture was analyzed via a texture profile analysis using a TA.XT2i Texture Analyzer (Stable Micro Systems, Surrey, UK) with a load cell of 25 kg and a Texture Expert for Windows version 1.2 Software. Each disc of dough was subjected to two cycles of compression (deformation = 40%, crosshead speed = 0.5 mm/s) with a cylindrical probe (P/75) [25]. The hardness (Hard), adhesiveness (Adh), cohesiveness (Cohes) and springiness (Sprin) of dough were calculated. Assays were performed in duplicate (*n* = 15).

#### 2.4.6. Dough Viscoelasticity

Measurements were performed in a HAAKE RheoStress 600 (Thermo Electron, Karlsruhe, Germany) at 25 ± 0.1 °C, using a serrated surface plate–plate sensor system (35 mm diameter) with 1.5 mm gap between plates. The upper plate was lowered, and the excess of sample was trimmed off. The exposed surface was covered with a thin layer of semisolid silicone to prevent moisture loss during the assay. Before testing, the samples were left for relaxation for 15 min. Shear stress sweep tests were previously performed to determine the linear viscoelastic range. Frequency sweeps (from 0.005 to 100 Hz) at constant stress (5 Pa), within the linear viscoelastic range, were performed [25]. Assays were performed in duplicate. 

Mechanical spectra were obtained by recording the dynamic moduli G′, G″ and tan δ as a function of frequency. The dynamic behavior of the dough was also analyzed by the complex modulus (G*) calculated according to Equation (2). In addition, G* as function of angular frequency (ω) was plotted and then the power law was fitted with Equation (3).
G* = √(G′)2 + (G″)2(2)
G* = AF × ω_1_/z(3)
where z represents the extension of the network, i.e., the number of flow units interacting each other (rheological units), and AF is interpreted as the dough strength, i.e., the intensity of those interactions between flow units [26].

### 2.5. Bread Quality Evaluation

Dough was prepared according to Section 2.4.2 but with 3% fresh yeast. Dough portions (50 g) were placed in individual cone aluminum molds (upper diameter = 65 mm, bottom diameter = 40 mm, height = 50 mm). Molds were placed for 80 min in a fermentation chamber (Brito Hnos, Buenos Aires, Argentina) at 30 °C. Baking was performed at 210 °C during 23 min in the oven (Ariston, Buenos Aires, Argentina). Breads were removed from the pans and cooled up to room temperature (2 h) before testing. Assays were performed by duplicate.

#### 2.5.1. Specific Volume

The specific volumes (Vs) were calculated as the ratio between volume and weight. Five breads of each formula were analyzed. 

#### 2.5.2. Protein and Moisture Content

The protein content and crumb moisture of breads were determined according to AACC Method [23]. Values obtained were the means of three replicates.

#### 2.5.3. Water Activity and Molecular Mobility

The water activity and molecular mobility of the different breadcrumbs were performed by the method described for the dough in Section 2.4.3 and Section 2.4.4, respectively.

#### 2.5.4. Texture Properties

Texture profile analysis of bread crumbs were undertaken in a TA.XT2i Texture Analyzer (Stable Micro Systems, Surrey, UK) using a load cell of 25 kg. Middle bread slices (thickness = 20 mm) underwent a double compression cycle (two-bite texture profile) up to 40% deformation of its original height with a disc probe (SMS/35). Eight replicates were analyzed for each kind of bread [25]. Firmness, springiness, resilience, cohesiveness and chewiness were obtained as textural parameters.

### 2.6. Statistical Analysis

Results were analyzed using Statgraphics Plus for Windows 5.1 software. Fisher’s least-significant differences test (*p* < 0.05) was used to define differences between means for each lupin flour. Means and standard deviations were calculated for each parameter.

## 3. Results and Discussion

### 3.1. Nutritional Composition and Thermal Stability of Proteins of White Lupine Flours

Rumbo and Misak lupine flours had a moisture content of 8.41% and 6.93%, while the protein content was 50.64% and 34.25%, respectively. In addition, Misak contained 2.80% ash, 55.7% carbohydrate and 11.6% of lipids; while Rumbo presented 3.64% ash, 27.2% carbohydrate and 10.14% lipids. The thermal behavior of the lupine flours was studied using differential scanning calorimetry (DSC). Figure 1 shows the thermograms obtained from the suspensions of the two flours. Three endothermic peaks were observed in both samples, with statistically similar onset, peak and end temperatures: the first at 77 °C was the smallest, the larger was at 88–89 °C and the other was between 104–105 °C.

Sousa et al. [27] previously studied the thermal stability of lupine (*Lupinus albus* L.) protein isolate by DSC, assigning the first peak at approximately 91 °C to the 2S globulin fraction, while the second and third peaks, at 100 °C and 106 °C, to the 7S and 11S globulins fraction, respectively. The higher temperatures reported could be due to the low amount of water used (1 g water/g solids). On the other hand, Muranyi et al. [28] studied the thermal properties of micellar lupin protein isolates in saline solution (MLP). The thermogram of MLP showed three endothermic transitions with peak temperatures of 87.37 °C and 101.2 °C and a minor peak at 70.0 °C, which were attributed to the denaturation of 7S (conglutins γ and β), 11S (conglutin α) and 2S globulins (conglutin δ), respectively. Mazumder et al. [29] studied, among other characteristics, the thermal properties of lupine flour suspensions of three Australian cultivars and found similar thermal behavior to that observed in our case. Following the same sequence, the imperceptible peak at 77–78 °C (Figure 1) would correspond to the 2S fraction, the peak at 88–89 °C to the 7S, while the highest temperature peak (104–105 °C) could be the 11S globulin fraction. 

On the other hand, the denaturation enthalpies of the protein fractions of the suspensions of Rumbo flour were statistically higher than those obtained for the Misak for both fractions (7S and 11S). This difference could be attributed to the high amount of proteins of Rumbo, which could be stabilized with a higher proportion of hydrogen bonds, and therefore require more energy to denature. Muranyi et al. [28] found significantly higher values of denaturation enthalpy values for both the 7S and 11S fractions with values of 9.4 and 7.5 J/g protein, respectively.

### 3.2. Hydration and Rheological Properties of Lupine Wheat Dough

Farinograms of blends of wheat flour with 10%, 20% and 30% lupine flours—Misak M (left) and Rumbo R (right)—are shown in Figure 2. Two peaks can be observed in all the farinograms; the first corresponds to the hydration of the components of the mixtures and the second to the increase in consistency due to the formation of the gluten network. No major differences were observed in the farinographic parameters of the blends prepared with the two lupin cultivars at the same addition level. In both systems, an increase in lupin flour increased the consistency of the second peak relative to the first by 50 BU (M10 and R10), by 160 BU in blends with 20% lupine (M20 and R20), and, in blends with 30% lupine (M30 and R30), the second peak was about 200 BU more consistent than the first one. The farinogram water absorption (Wabs) of wheat flour was 60% and the development time was 19 min (farinogram not shown). An increase in Misak flour (≥20%) increased W_abs_, whereas mixtures with Rumbo variety absorbed 10% less water than wheat flour. Regarding the development time (td), this parameter decreased in both cases. The decrease was more pronounced in the Misak variety. Finally, the stability (St) decreased in both systems, while the degree of softening (SD) increased.

Dervas et al. [30] reported an increase in W_abs_ when 5, 10 and 15% of *Lupino albus* spp. flour or defatted flour was added to wheat flour. Similar behavior has been reported with the incorporation of other legumes into wheat flour and an increase in W_abs_ with carob germ flour [25], pea or lupin proteins isolates [31]. These authors attributed this behavior to the protein content as well as to the technofunctional properties of lupine proteins (globular proteins), such as high water and oil absorption and foaming and emulsifying capacity, compared to soy protein.

On the other hand, Piasecka-Jóźwiak et al. [32] reported a decrease in dough development time and stability when replacing up to 25% of wheat flour with *Lupinus luteus* (34.6% protein, d.b.) and *Lupinus angustifolius* (25.2% protein, d.b.). These authors attributed the decrease in stability to the high content of legume proteins, obviously less marked in *L. angustifolius*, leading to a dilution of the gluten proteins. In a recent work, Carboni et al. [33] demonstrated the importance of performing a farinographic test on blends containing wheat and legume flours to ensure good baking performance.

Incorporation of the two lupine flours, Misak and Rumbo, resulted in earlier gluten network formation, although wheat proteins were diluted, and stability decreased between 82 and 90% in blends with 30% compared to 10% lupine flour addition. However, both flours contain significant amounts of globular proteins, which seem to favor the formation of the gluten network, although this reinforcement is not maintained during kneading. Therefore, the technological processes that should be performed during baking could have a negative impact on the quality of the product.

The moisture (M_cont_), water activity (a_w_) and molecular mobility (λ) of the different doughs are shown in Table 1. Doughs with 10 and 20% of Misak flour (M10 and M20) presented lower moisture content, while the moisture of the M30 dough was higher than that of the wheat dough (C). With the addition of Rumbo lupine flour, a decrease in moisture content was observed in all the dough with respect to the C, which was more marked in the case of R10 and R30. The water content of the different doughs correlated with the farinographic absorption of the mixtures. Although moisture content varied for the different formulations, no significant differences were observed in the aw values (Table 1).

The molecular mobility of wheat dough (C) was the highest. An increase in lupine flours, Misak and Rumbo, λ parameter significantly decreased. This trend suggests that the molecular mobility is lower and therefore a more rigid matrix would be formed due to the incorporation of different lupine flour components, especially the globular proteins. However, at the same level of lupine flour addition, the values were even significantly lower in Rumbo dough than in Misak dough (Table 1). This behavior could be due to the fact that the Rumbo cultivar contains 50% protein and, in turn, required less water to form a dough of the desired consistency, so that both types of proteins could be responsible for a more structured, i.e., more solid or rigid, matrix. The parameters obtained from the texture profile analysis are shown in Table 2. Dough with 10 and 20% of Misak (M10 and M20) significantly increased hardness, adhesiveness and cohesiveness, and when 30% of this flour was added (M30), these parameters decreased, reaching values significantly lower than those obtained for wheat dough (C). No significant differences in springiness were observed with the addition of Misak flour. In fact, M30 is not really a dough, resembles more a paste of lower elasticity than dough, in concordance with high relaxation (very low λ) and very low hardness. On the other hand, in the dough with Rumbo, hardness and adhesiveness significantly decreased with increasing amounts of this flour in the mixture, while cohesiveness and elasticity decreased, except for R20, which presented higher and equal values than wheat dough, respectively.

In summary, while dough hardness and adhesiveness increased with Misak (up to 20%), with Rumbo they decreased to values close to that of wheat dough, and values with Rumbo were much lower than those observed of Misak dough at the same substitution. The difference in the textural behavior of dough with the different lupine flours could be due to the distinct profile and protein content of both lupine varieties. Salinas et al. [25], who studied the effect of the incorporation of carob germen wheat flour on the rheological properties of the dough, found that blends with 30% of carob increased its hardness. In addition, other authors found that an increase in this parameter was also found in wheat dough supplemented with other protein sources, such as amaranth (up to 30%). These authors attributed this behavior to the fact that proteins absorb water, gelation occurs, increasing consistency and structuring the dough matrix [34]. Although Misak has a lower protein content, it would stabilize the gluten network to a greater extent than Rumbo, probably forming a gel structure in interaction with water, which would have an inhibitory effect on the dilution of the gluten-forming wheat proteins. With 30% of lupine flours, the gluten is highly diluted, and it would not be possible to develop a matrix with adequate characteristics, resulting in a less cohesive and elastic dough. There were no significant differences in the values of elastic modulus G′ or tan δ (≅0.4), measured at 1 Hz, between the different formulations studied (Table 2). Ahmed et al. [35] and Liu et al. [36] obtained slightly different results to the present work. They observed the highest value of G′ for wheat dough, and values decreased with increasing amounts (5–30%) of lupine incorporation. In addition, these authors found that the tan δ value was the lowest for the wheat dough and increased with increasing amounts of lupine; consequently, the dough became more viscous (tan δ = G″/G′).

The variation of the viscoelasticity of the dough at different frequencies with the addition of lupine flour is shown in Figure 3. Figure 3a,b shows the relationship between G′ and G″ in Misak and Rumbo doughs, respectively. All the doughs with Misak showed the same variation, with no differences in flour level with respect to the control wheat dough. All the doughs were above the straight line at 45°, indicating that G′ was greater than G″, and with curves practically parallel to this straight line; therefore, a homogeneous matrix was formed in all the dough.

On the one hand, in the case of doughs with Rumbo, although they showed a similar behavior to the Misak dough, there were some differences. The R20 dough behaved like C dough, with curves above those of R10 and R30, which were also similar. This slight superiority in the curves of C and R20, observed up to values of about 8000 Pa of G″, implies a higher value of G′ for the same value of G″ at low frequencies (ω < 0.7 Hz). On the other hand, dough R10, and especially dough R30, presented at frequencies lower than 0.07 Hz; higher values of G″ form the same value of G′, suggesting the formation of less elastic and more viscous dough. This behavior was different to that observed in dough with amaranth flour (pseudocereals) [34], probably due to the different kind and composition of these proteins and therefore to their relationship with wheat proteins during gluten formation.

Another way for studying the dynamic viscoelasticity of dough is through the variation of G* with angular frequency (Figure 3c,d). Dough with Rumbo flour behaved differently than dough with Misak. Again, R20 behaved like C, while R10 and R30 (Figure 3d) presented G* values higher than R20, as well as M10 and M30 (Figure 3c). In the power law, Arp et al. [25] related z to the number of flow units (rheological units) that interact each other, and AF to the intensity of those interactions related to dough strength. These authors found a value of z of 5.97 in the absence of hydrocolloids, which decreased to 4.86 and 4.80 for dough with hydroxypropyl methylcellulose F4M and carboxymethylcellulose, respectively. In our case, the values of AF and z of dough did not significant change with the addition of Misak flour, while for Rumbo sample’s AF increased, and z decreased significantly with the addition of 30% lupine flour compared to the wheat dough. Results suggest that for R30, a low number of interacting units (z) were formed in dough but with high intensity in their interactions (AF), suggesting the formation of a network of high strength interactions. These high-strength bonds could be stabilized by the globular proteins present in Rumbo lupine flour. Although R30 is a highly elastic dough (high G* due to low z and high A_F_), these bonds between the polymers are easily broken during compression (TPA test), resulting in a softer dough after this process. In the case of dough with 30% of resistant starch and hydrocolloids (up to 1.5%), Arp et al. [25] found that a wheat dough is constituted by flow units that are interacting with each other in a cooperative three-dimensional arrangement instead of by the single breakable strands of the strong gel model. In the case of lupine wheat dough, lupine proteins produce in R30 a single breakable strand, resulting in a dough with low hardness and cohesiveness.

### 3.3. Bread Quality of Wheat Bread Complemented with Lupine

Quality parameters of bread (specific volume of bread and protein content) and bread crumbs (moisture content, molecular mobility, and water activity) of lupine wheat bread are described in Table 3.

These results contrast with those found by other authors who used lupine flour in different proportions. They found that the Vs of the breads was lower than that of wheat bread, and furthermore as the amount of lupine flour increased, the bread volume decreased [31,36,37,38]. Baking quality was negatively affected when lupin flour was used as a nutritional ingredient in organic wheat sourdough bread [33]. In turn, as expected, as the percentage of both Misak and Rumbo flour increased, protein content increased proportionally with the higher percentage of lupine flour added. However, the values for breads with Rumbo flour were higher than those with Misak flour, due to the higher protein content of this flour (Table 3). Several authors found that the incorporation of lupine flour in breads, in various proportions, resulted in an increase in protein content compared to wheat bread [33,39,40].

The moisture of the breads with Misak flour decreased at the lowest percentages (10 and 20%); however, at 30%, it had a moisture similar to that of wheat bread (C). Meanwhile, the breads made with Rumbo flour showed values lower than C in all additions, with the lowest value being for R30. Although the water content varied in both types of bread, the water activity remained constant in all breads. These results suggest that Rumbo lupin globular proteins may form a bread matrix together with gluten with a lower proportion of water, in agreement with the lower farinographic water absorption obtained for the R30 blend (Figure 1).

In addition, when observing the λ value in the breads with Misak lupine flour, a significant decrease is observed with increasing flour content, associated with a more flexible matrix. In breads with Rumbo incorporation a decrease in molecular mobility was observed in all cases with respect to wheat bread, suggesting that Rumbo proteins form a more compact matrix with less relaxation.

A slice of wheat bread containing lupine flours (Misak and Rumbo) is shown in Figure 4.

Finally, the textural parameters of the breads are shown in Figure 5. Firmness and chewiness increased with increasing lupine flour content for both Misak and Rumbo. However, it is important to note that replacement with Rumbo flour up to 20% did not change crumb firmness; it increased in R30 breads, although, this increase was significantly less than for M30. This suggests that lupin breads with better textural quality can be obtained with Rumbo flour. A significant decrease in cohesiveness was observed with 10% Misak flour in the bread, with no differences between the other levels; however, in the breads with Rumbo flour, although there were no significant differences in cohesiveness, R10 and R20 showed a cohesiveness tending towards higher values. Crumb elasticity decreased as the amount of lupine flour increased for both types.

Several authors found that by incorporating lupine flour in wheat flour-based breads, the firmness of the breads was higher than in wheat bread, and that the firmness increased with the higher amount of lupine flour in the blend [30,32,36,39]. Hoehnel et al. [31] attributed the high firmness of breads to a competition for water between protein gelation and starch gelatinization during baking. If less starch gelatinizes due to protein gelation, a high amount of native starch would contribute to high hardness. In addition, Martínez et al. [41] reported the hydration depletion phenomenon of starch that would increase crumb hardness. Paraskevopoulou et al. [42] proposed the thickening of the crumb walls surrounding the air cells and the strengthening of the crumb structure by the protein particles as the common phenomenon for the increase in crumb hardness. In our case, the high firmness of breads made with Misak flour could be attributed to these two-phenomena proposed by these researchers. In addition, Misak flour exhibited the lowest denaturation enthalpy values for the 7S and 11S fractions, suggesting that Misak flour contained less amount of these proteins or that they were previously denatured. Denaturation of Rumbo lupine proteins would be stated during baking; these proteins could unfold at the solid–air interface, leading to a better dough expansion during the production of carbon dioxide, resulting in a softer crumb in a slightly-high-volume bread.

## 4. Conclusions

Despite the two lupine flours being from the same specie (*Lupinus albus* L.), they are from different cultivars and origins (one cultivated in America and the other in Europe), and conferred different characteristics, especially in the protein content and profile, which led to a different interaction with wheat flour and with water. This led to the formation of doughs with different hydration and rheological properties, which, in turn, led to breads of different baking quality. Although both varieties are feasible to be used in breads, the better technological quality of the Rumbo lupine flour obtained from Chile is evident.

The farinograms of the mixtures with both lupine flours were affected with supplementation ≥20%. The rheological properties of the dough were negatively affected with both lupine flours, especially with the Rumbo cultivar, where the hardness decreased significantly, and the intensity of the interactions was affected especially with 30% incorporation. Despite this, the breads presented good specific volumes. In addition, the firmness and chewiness of the crumbs increased, but less so in the breads with the Rumbo cultivar. Finally, breads with 30% of this lupine cultivar had the highest protein content. Therefore, even though both cultivars provide protein and reveal dough handling difficulties, the Rumbo cultivar is the one that affects the bread quality to a lesser degree.

The data obtained in the present study highlight the relevance of lupine flour as a high-nutrition ingredient for functional bakery products due to its suitable technological performance. Further research is needed to evaluate consumer acceptance of bread or baking products containing the commercial lupin varieties studied in the present work.

## Figures and Tables

**Figure 1 foods-12-01645-f001:**
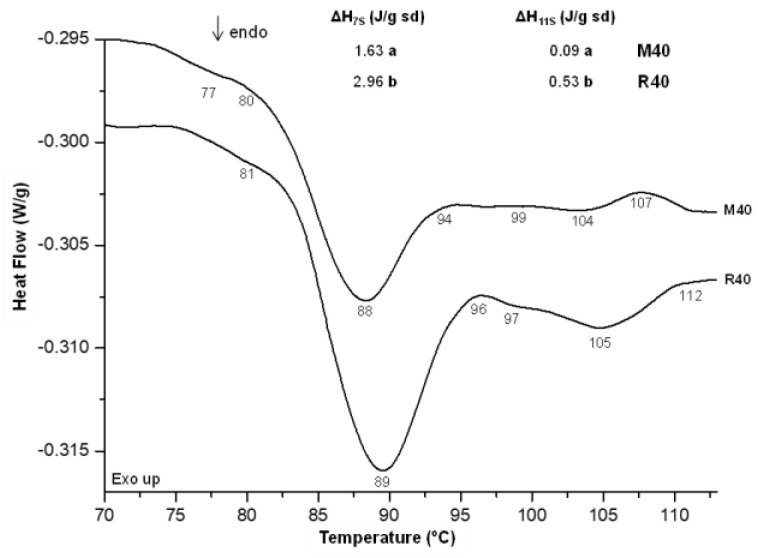
Differential scanning calorimetry thermograms of Misak and Rumbo lupine flour aqueous suspensions. Different letters in the same parameters indicate a significant difference (*p* < 0.05).

**Figure 2 foods-12-01645-f002:**
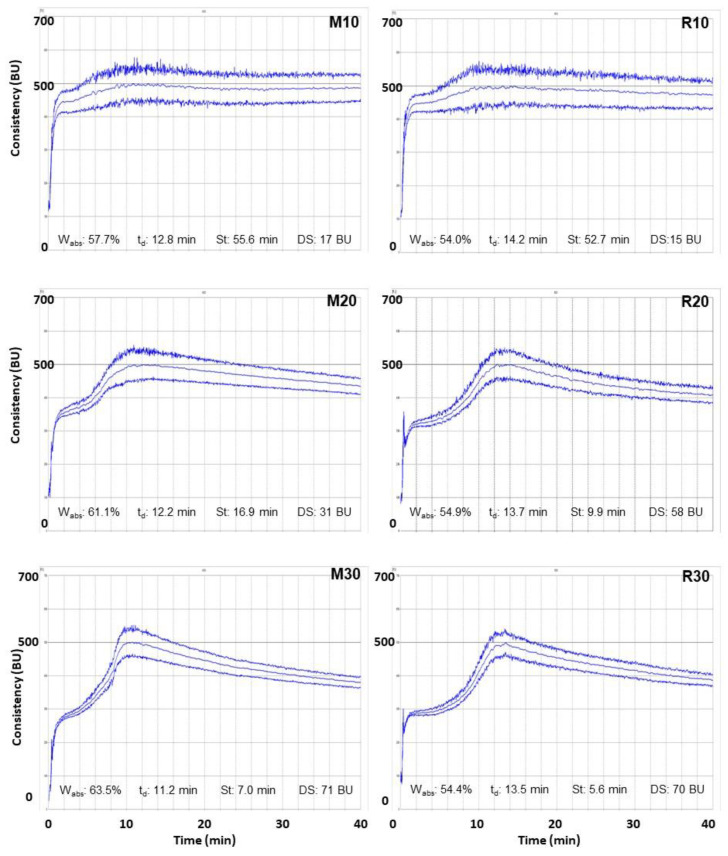
Farinograms of wheat flour complemented with Misak lupine flour: 10% (M10), 20% (M20) and 30% (M30); and with Rumbo lupine flour: 10% (R10), 20% (R20) and 30% (R30). W_abs_: water absorption; td: development time; St: stability; SD: softening degree. The *Y*-axis corresponds to the consistency (between 0 and 700 BU)’’ the marked line parallel to *X*-axis indicates 500 BU; the *X*-axis is the kneading time (up to 40 min) for all the graphs.

**Figure 3 foods-12-01645-f003:**
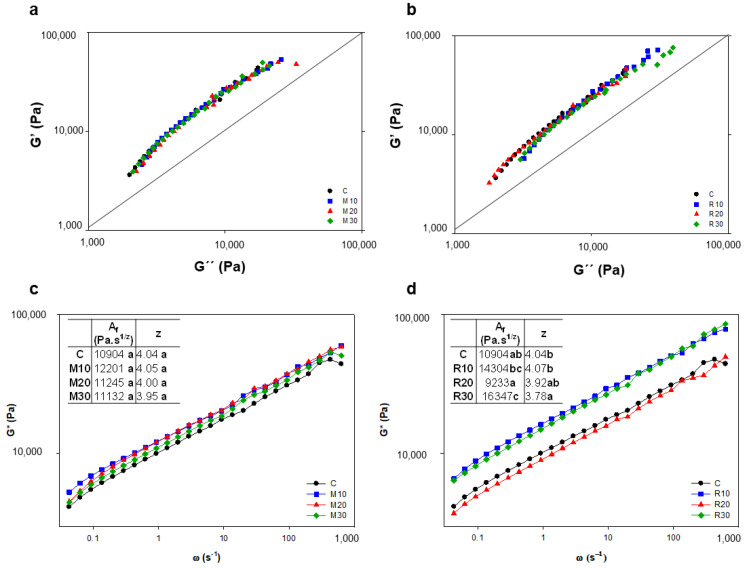
Small amplitude oscillatory rheology of dough. Elastic modulus (G′) as a function of viscous modulus (G″) of wheat flour dough with: Misak (**a**) and Rumbo (**b**) lupine flours. Complex modulus (G*) as a function of ω of wheat flour dough with Misak (**c**) or Rumbo (**d**) lupine flours. C: wheat dough. Wheat flour complemented with Misak: 10% (M10), 20% (M20) and 30% (M30); and with Rumbo: 10% (R10), 20% (R20) and 30% (R30). Different letters in the same parameters indicate a significant difference (*p* < 0.05).

**Figure 4 foods-12-01645-f004:**
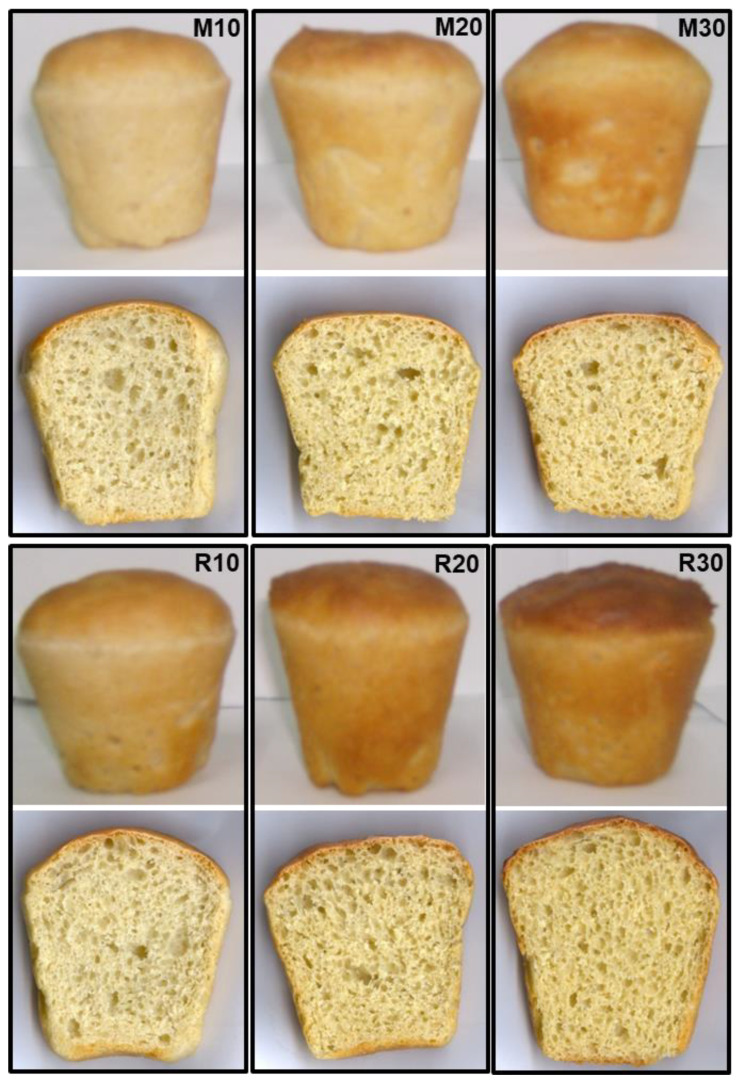
Slices of wheat-bread with incorporations of Rumbo flour (R10, R20 and R30) and Misak flour (M10, M20 and M30).

**Figure 5 foods-12-01645-f005:**
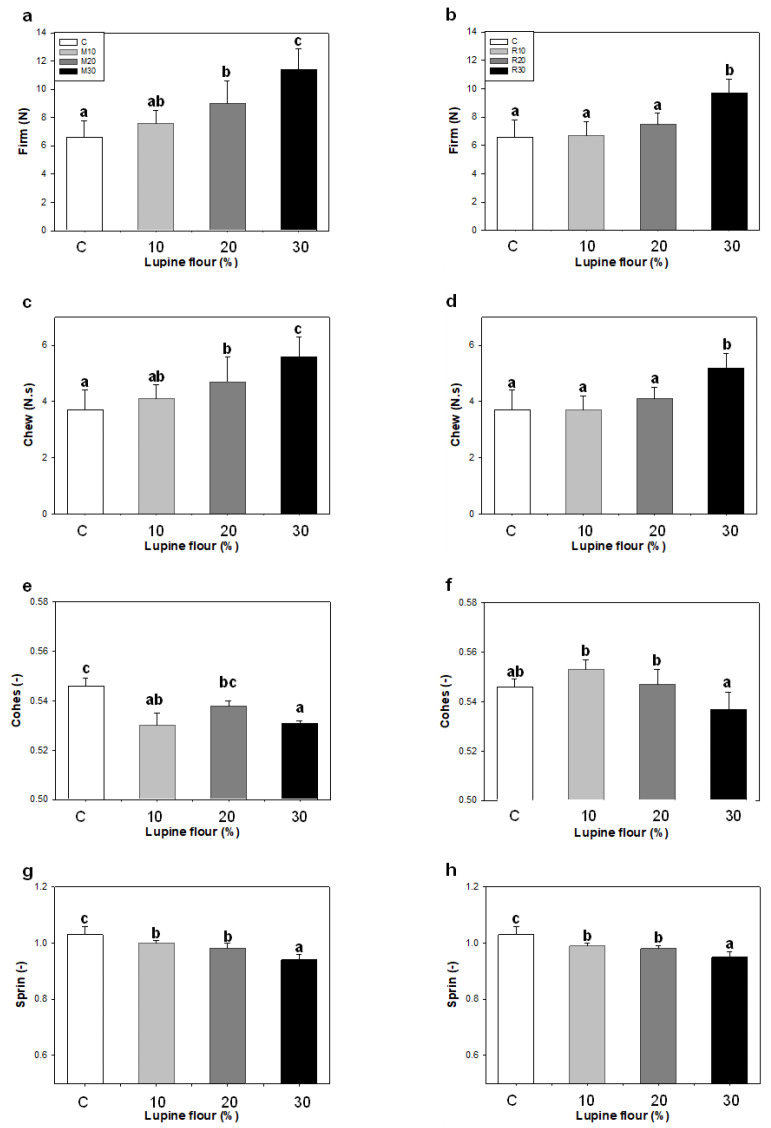
Texture parameters of bread crumbs with lupine flours: (**a**,**b**) Firm: firmness; (**c**,**d**) Chew: chewiness; (**e**,**f**) Cohes: cohesiveness; (**g**,**h**) Sprin: springiness or elasticity. Breads with Misak flour: (**a**,**c**,**e**,**g**). Breads with Rumbo flour: (**b**,**d**,**f**,**h**). Errors bars: standard deviations. Different letters indicate significant differences (*p* < 0.05).

**Table 1 foods-12-01645-t001:** Hydration parameters of lupine wheat flour dough.

Lupine Flour (%)		M_cont_ (%)	a_w_ (-)	λ (ms)
C	0	44.7 ± 0.2 c	0.973 ± 0.000 a	8.8 ± 0.9 d
M10	10	43.7 ± 0.5 a	0.971 ± 0.001 a	4.2 ± 0.1 c
M20	20	44.3 ± 0.0 b	0.973 ± 0.003 a	3.1 ± 0.4 b
M30	30	45.8 ± 0.1 d	0.973 ± 0.001 a	2.1 ± 0.3 a
C	0	44.7 ± 0.2 c	0.973 ± 0.000 b	8.8 ± 0.9 c
R10	10	42.9 ± 0.1 a	0.968 ± 0.001 a	2.3 ± 0.9 b
R20	20	43.2 ± 0.0 b	0.967 ± 0.002 a	1.3 ± 0.7 ab
R30	30	43.0 ± 0.1 a	0.967 ± 0.001 a	0.4 ± 0.1 a

Moisture content (M_cont_); water activity (a_w_); molecular mobility (λ). Each lupine flour’s values within a column followed by the same letters are significantly different (*p* < 0.05).

**Table 2 foods-12-01645-t002:** Rheological parameters of lupine wheat flour dough.

Lupine Flour (%)		Hard (N)	Adh (N.s)	Cohes (-)	Sprin (-)	G’ (kPa)	tan δ (-)
C	0	1.2 ± 0.1 b	4.3 ± 0.8 b	0.78 ± 0.03 a	0.91 ± 0.01 a	18 ± 3 a	0.41 ± 0.02 a
M10	10	6.0 ± 1.3 d	18.1 ± 2.8 d	0.86 ± 0.04 b	0.91 ± 0.01 a	16 ± 1 a	0.38 ± 0.02 a
M20	20	4.0 ± 0.6 c	15.1 ± 2.3 c	0.87 ± 0.04 b	0.89 ± 0.01 a	17 ± 1 a	0.40 ± 0.02 a
M30	30	0.8 ± 0.1 a	3.3 ± 0.4 a	0.80 ± 0.03 a	0.89 ± 0.02 a	16 ± 1 a	0.39 ± 0.00 a
C	0	1.2 ± 0.1 c	4.3 ± 0.8 c	0.78 ± 0.03 ab	0.91 ± 0.01 b	18 ± 3 ab	0.41 ± 0.02 a
R10	10	1.0 ± 0.1 b	4.4 ± 0.3 c	0.75 ± 0.02 a	0.89 ± 0.01 ab	21 ± 4 ab	0.42 ± 0.01 a
R20	20	0.9 ± 0.2 b	3.6 ± 0.7 b	0.82 ± 0.05 b	0.91 ± 0.02 b	14 ± 1 a	0.41 ± 0.02 a
R30	30	0.6 ± 0.1 a	2.4 ± 0.4 a	0.75 ± 0.03 a	0.87 ± 0.02 a	24 ± 3 b	0.44 ± 0.02 a

Textural parameters: Hardness (Hard); Adhesiveness (Adh); Cohesiveness (Cohes); Springiness (Sprin). Dynamics rheological parameters: Storage modulus (G′) and tan δ (G″/G′) at ω = 1 Hz. Each lupine flour’s values within a column followed by the same letters are significantly different (*p* < 0.05).

**Table 3 foods-12-01645-t003:** Quality parameters of lupine wheat bread.

	Lupine Flour (%)	Bread		Bread Crumb		
		Vs (cm^3^/g)	Protein (%)	M_cont_ (%)	λ (ms)	a_w_ (-)
C	0	2.5 ± 0.2 a	7.27 ± 0.00 a	46.4 ± 0.1 c	6.8 ± 0.2 d	0.966 ± 0.000 a
M10	10	2.6 ± 0.2 a	8.69 ± 0.11 b	44.5 ± 0.2 a	4.2 ± 0.3 c	0.968 ± 0.003 a
M20	20	2.6 ± 0.1 a	9.08 ± 0.10 c	45.5 ± 0.1 b	3.5 ± 0.2 b	0.970 ± 0.001 a
M30	30	2.4 ± 0.1 a	9.59 ± 0.14 d	46.2 ± 0.0 c	3.0 ± 0.3 a	0.967 ± 0.001 a
C	0	2.5 ± 0.2 a	7.27 ± 0.00 a	46.4 ± 0.1 c	6.8 ± 0.2 c	0.966 ± 0.000 b
R10	10	2.9 ± 0.1 b	9.30 ± 0.04 b	43.7 ± 0.1 b	2.3 ± 0.6 ab	0.963 ± 0.000 ab
R20	20	2.6 ± 0.2 a	10.68 ± 0.15 c	43.6 ± 0.1 b	1.8 ± 0.7 a	0.965 ± 0.001 b
R30	30	2.5 ± 0.2 a	13.55 ± 0.06 d	43.5 ± 0.0 a	3.0 ± 0.3 b	0.966 ± 0.000 b

Parameters: specific volume of bread (Vs); protein percentage; moisture content (M_cont_); molecular mobility (λ); and water activity (a_w_). For each lupine flour, values within a column followed by the same letter are not significantly different (*p* < 0.05).

## Data Availability

Data presented in this study are available on request from the corresponding author.

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
