# Peer review of "White Lupine (Lupinus albus L.) Flours for Healthy Wheat Breads: Rheological Properties of Dough and the Bread Quality"

_foods, 2023, doi:10.3390/foods12081645_

Round 1
Reviewer 1 Report
This work presents Assessment of the incidence of white lupines (Lupinus albus L.) flours on rheological properties of wheat dough and the bread quality. This manuscript (MS) title is interesting; however, I have some general observation before going in depth revision, authors are encouraged to modify the MS and address the issue before final decision.
The authors should clearly state the research question and hypothesis to guide the study.
1. As this study relevant to food production, the question is why such study was need? Is any market demand or consumer preference? Authors must mention this with some statistically available figures.
2. Improve the methodology: The authors should provide a more detailed description of the experimental design, including the sample size and statistical analysis used.
3. It would also be helpful to provide information on the type of wheat used and the level of lupine flour added to the dough.
4. The authors should provide more detailed information on the results of the study, including data on the specific rheological properties of the dough and the bread quality.
5. It would also be helpful to include pictures of the bread to illustrate the differences in quality.
6. The authors should acknowledge the limitations of the study and provide suggestions for future research to address these limitations.
7. The authors should improve the clarity and organization of the paper, including the use of headings and subheadings to guide the reader.
8. Overall, the authors should provide more details and clarity in their paper to better support their conclusions.
9. In last the Abstract need revision and must be different from conclusion
Author Response
Dear reviewer
Authors are very grateful to the reviewers for the contributions to improve our work. The manuscript has been changed, taking all the reviewers’ remarks into account.
This work presents Assessment of the incidence of white lupines (Lupinus albus L.) flours on rheological properties of wheat dough and the bread quality. This manuscript (MS) title is interesting; however, I have some general observation before going in depth revision, authors are encouraged to modify the MS and address the issue before final decision.
The authors should clearly state the research question and hypothesis to guide the study.
- As this study relevant to food production, the question is why such study was need? Is any market demand or consumer preference? Authors must mention this with some statistically available figures.
Response: In the introduction section, information about worldwide demand for breads has been incorporated.
- Improve the methodology: The authors should provide a more detailed description of the experimental design, including the sample size and statistical analysis used.
Response: In the methods section information about experimental design was clarified together with “statistical analysis”. All assays were performed by duplicate.
- It would also be helpful to provide information on the type of wheat used and the level of lupine flour added to the dough.
Response: The type of wheat flour used was type 65 (Portugal) and type 0000 (according to Argentinean legislation), and the levels of lupine flour added to 100 g of wheat flour were 10, 20 and 30%. This information was added and mentioned in Sub-section 2.4.
- The authors should provide more detailed information on the results of the study, including data on the specific rheological properties of the dough and the bread quality.
Response: Detailed information about rheological properties of dough and bread quality were previously described. Parameters of Table 2 and Figure 3, belonging to texture and viscoelasticity of dough, were widely analysed. In addition, parameters of Table 3 belonging to the volume and protein content of breads and also moisture, water activity and molecular mobility of crumbs, were deeply described. Figure 4 shows the form of the bread pieces, like “muffins” and slices of breads where it can be seen crumbs structures and description and analysis of TPA parameters placed in Figure 5 were complete.
- It would also be helpful to include pictures of the bread to illustrate the differences in quality.
Response: Pictures of the different breads and their crumbs were included in a reformulated Figure 4.
- The authors should acknowledge the limitations of the study and provide suggestions for future research to address these limitations.
Response: limitations of the study were considered and suggestions for future research with these limitations were included in the Conclusion section.
- The authors should improve the clarity and organization of the paper, including the use of headings and subheadings to guide the reader.
- Overall, the authors should provide more details and clarity in their paper to better support their conclusions.
- In last the Abstract need revision and must be different from conclusion
Response: All manuscript was revised and reformulated.
Reviewer 2 Report
This is a good paper and a very interesting study. The English is poor in many places, e.g. line 50, page 2, “. Other fraction is albumin that reaches values of 15% of…” (odd expression).
The experimentation is very competent and appropriate experimentation, e.g. dough rheology. There is good comparison with the literature, with very relevant references, and the statistical analysis and testing is appropriate. The discussion of the different proteins and their degradation processes is very relevant and well done.
Author Response
Dear reviewer
Authors are very grateful to the reviewers for the contributions to improve our work. The manuscript has been changed, taking all the reviewers’ remarks into account.
-This is a good paper and a very interesting study. The English is poor in many places, e.g. line 50, page 2, “. Other fraction is albumin that reaches values of 15% of…” (odd expression).
Response: English grammar was corrected. Redaction related to albumin fraction was also corrected.
The experimentation is very competent and appropriate experimentation, e.g. dough rheology. There is good comparison with the literature, with very relevant references, and the statistical analysis and testing is appropriate. The discussion of the different proteins and their degradation processes is very relevant and well done.
Reviewer 3 Report
Revision required
Please review the attached file.

Author Response
Dear reviewer
Authors are very grateful to the reviewers for the contributions to improve our work. The manuscript has been changed, taking all the reviewers’ remarks into account.
The present study investigated the incidence of white lupines (Lupinus albus L.) flours on rheological properties of wheat dough and the bread quality. There are several communication problems: from English at times difficult to understand, to the confusion about the terms and concepts used. To improve the quality of the article, the following are suggested:
- The article as well as the title have been drafted without a clear rationale
Response: the title and some discussions of the manuscript were improved with the aim of clarifying the meaning of the work.
- Line 25-26, rephrase it please
Response: The sentence was reformulated.
- Abstract needs a lot of attention and should cover theme of whole manuscript
Response: the manuscript, as it was also suggested by the others reviewers, was revised and reformulated.
- Line 37, there should be space between reference and text
Response: the space between reference and text was added.
- Line 77-80, authors did all the analyses. Otherwise add references?
Response: results of composition of lupine flours were moved to Results and Discussion section, as it was suggested by other Reviewers. A new section was included with the corresponding reference AACC methods [20]. (2.2. Percentage Composition of lupines flours).
- There is repetition of data in lines 78-79 and 200-201
Response: The nutritional composition of the lupines flours that was carried out in our laboratory, was eliminated form lines 78-79 and moved to section 3.1. (3.1. Nutritional composition and thermal stability of proteins of white lupine flours).
- Line 90, 108, 141, 150, 190 add appropriate reference?
Response: appropriate references were included in all Materials and Methods sections.
- Figure 2 legends are not visible, please improve the quality of images or write the legends manually if possible?
Response: The X-axis legend was added. In general, the farinographam is centred on 500 BU, this was clarified in figure caption.
- Any reference to support the argument authors have given in lines 326-330
Response: a reference was included in the text.
- The abstract should focus on the findings of the manuscript.
Response: the abstract was improved according to Reviewer suggestion.
- A conclusive line should also be added at the end of abstract.
Response: a conclusive line was included at the end of the manuscript.
- Spelling errors were observed in some instants
- Grammatical errors in several places
Response: spelling and grammatical errors were checked and corrected.
- Summarize updated recent research related to the topic
Response: a recent research related to the topic was included in the text.
- Highlight gaps in current understanding or conflicts in current knowledge
- Establish the originality of the research aims by demonstrating the need for investigations in the topic area
Response: our work gave responses to the current knowledge of wheat-legume blends for breadmaking, specifically with two varieties of highly consume legume as white lupine. We found that differences in the precedence of the variety will affect dough properties and its performance during baking, leading to breads of different technological and nutritional quality. Therefore, it is an original work on functional breads formulated with white lupine. Legumes present each one their particularity that will determine their behaviour when they are included as raw ingredients for foods. In the case of breads, dough should present certain characteristics, which are needed to well described, for assuring high-quality bread.
- Only use scientific words throughout the manuscript
Response: scientific words were used throughout the manuscript.
Reviewer 4 Report
The objective of this paper is to study the thermal properties of two lupine flours (Misak and Rumbo), and the influence of different amounts of lupine flours (0%, 10%, 20% and 30%) incorporations on the hydration and rheological properties of dough and bread quality. The subject is interesting because of to the potential of using lupine flours in food industry due their technological characteristics and exceptional nutritional value. The tables and figures are clear and the conclusion are adjunsted to the results obtained however there are some modifications needed:
The introduction should include some information about the allergenic properties of lupins and the treatments to reduce it . Some references concerning this are: Alvarez et al 2005 doi10.1021/jf0490145;
The introduction also should contain previous information about breadmakin properties of adding lupins. The reference to the work of Guillamon et al 2010 ( doi:10.5424/sjar/2010081-1148) reporting that thermally-treated lupine flours, had similar breadmaking and sensorial properties as untreated lupine flour. and these thermal treatments could increase the potential use of lupine flour as a food ingredient while reducing the risk to provoke allergic reactions, should be included in the introduction and discussion.
In my opinion a sensory analysis of the bread produced with lupins (both varieties) should be carried out and the results compared with other previous.
In figure 4 caption, the last part is repeaded.
Author Response
Dear reviewer
Authors are very grateful to the reviewers for the contributions to improve our work. The manuscript has been changed, taking all the reviewers’ remarks into account.
The objective of this paper is to study the thermal properties of two lupine flours (Misak and Rumbo), and the influence of different amounts of lupine flours (0%, 10%, 20% and 30%) incorporations on the hydration and rheological properties of dough and bread quality. The subject is interesting because of to the potential of using lupine flours in food industry due their technological characteristics and exceptional nutritional value. The tables and figures are clear and the conclusion are adjunsted to the results obtained however there are some modifications needed:
-The introduction should include some information about the allergenic properties of lupins and the treatments to reduce it . Some references concerning this are: Alvarez et al 2005 doi10.1021/jf0490145
Response: information about the allergenic properties of lupins and the treatments to reduce them, were included. Also, the reference of Alvarez et al 2005 was considered.
Álvarez-Álvarez, J., Guillamón, E., Crespo, J. F., Cuadrado, C., Burbano, C., Rodríguez, J., ... & Muzquiz, M. (2005). Effects of extrusion, boiling, autoclaving, and microwave heating on lupine allergenicity. Journal of Agricultural and Food Chemistry, 53(4), 1294-1298.
-The introduction also should contain previous information about breadmakin properties of adding lupins. The reference to the work of Guillamon et al 2010 ( doi:10.5424/sjar/2010081-1148) reporting that thermally-treated lupine flours, had similar breadmaking and sensorial properties as untreated lupine flour. and these thermal treatments could increase the potential use of lupine flour as a food ingredient while reducing the risk to provoke allergic reactions, should be included in the introduction and discussion.
Response: information about breadmaking properties of lupins was included in the Introduction section. The reference Guillamon et al 2010 was incorporated.
Guillamón Fernández, E., Cuadrado Hoyos, M. C., Pedrosa, M. M., Varela, A., Cabellos Caballero, B. N., Muzquiz, M., & Burbano, C. (2010). Breadmaking properties of wheat flour supplemented with thermally processed hypoallergenic lupine flour.
-In my opinion a sensory analysis of the bread produced with lupins (both varieties) should be carried out and the results compared with other previous.
Response: we agree with the Reviewer opinion, but unfortunately it was not possible to perform the sensory analysis of wheat-lupine breads formulated at the present work. However, for further research we expected to evaluate the consumer’s acceptance of lupine-wheat breads.
-In figure 4 caption, the last part is repeaded.
Response: the last part of Figure 4 caption was removed.
Round 2
Reviewer 1 Report
no further comments. accepted in current position.
Reviewer 4 Report
The authors have reviewed the manuscript according my previous suggestions and comments. They answered satisfactorily previous questions possed. So, I recommend to accept this modified version